# The Impact of Discrete Element Method Parameters on Realistic Representation of Spherical Particles in a Packed Bed

Zahra Ghasemi Monfared *, J. Gunnar I. Hellström 🔟 and Kentaro Umeki 🔟

Department of Engineering Sciences and Mathematics, Luleå University of Technology, 971 87 Luleå, Sweden; gunnar.hellstrom@ltu.se (J.G.I.H.); kentaro.umeki@ltu.se (K.U.)
* Correspondence: zahgha@ltu.se

**Abstract:** Packed bed reactors play a crucial role in various industrial applications. This paper utilizes the Discrete Element Method (DEM), an efficient numerical technique for simulating the behavior of packed beds of particles as discrete phases. The focus is on generating densely packed particle beds. To ensure the model accuracy, specific DEM parameters were studied, including sub-step and rolling resistance. The analysis of the packed bed model extended to a detailed exploration of void fraction distribution along radial and vertical directions, considering the impact of wall interactions. Three different samples, spanning particle sizes from 0.3 mm to 6 mm, were used. Results indicated that the number of sub-steps significantly influences void fraction precision, a key criterion for comparing simulations with experimental results. Additionally, the study found that both loosely and densely packed beds of particles could be accurately represented by incorporating appropriate values for rolling friction. This value serves as an indicator of both inter-particle friction and friction between particles and the walls. An optimal rolling friction coefficient has been thereby suggested for the precise representation for the densely packed bed of spherical char particles.

**Keywords:** packed bed; discrete element method; rolling friction; void fraction; sub-steps; wall effect

## 1. Introduction

Packed beds, also called fixed beds, of uniform and non-uniform solid particles have been widely utilized in chemical and process industries in various particle-based applications, from catalytic reactions and wastewater treatment to gasification/pyrolysis, iron making, absorption/desorption, cooling, and energy storage. Packed beds are randomly filled beds of particles with porous structures that ensure low flow resistance while enhancing heat and mass transfer due to a high surface area-to-volume ratio [1]. To maximize the advantages, it is crucial to design packed beds with a detailed understanding of the porous bed structure and its effect on the transport phenomena. In this regard, CFD (Computational Fluid Dynamics) simulations have come to help experimental analysis [2]. However, the conventional porous media approach does not consider the solid phase as discrete particles and requires models for void fraction and transport parameters to describe the effect of particle packing [3], i.e., continuum approach. Therefore, prior to CFD simulations, a realistic packed bed of particles should be generated in a way that gives detailed information about the nature of the bed and the arrangements of the particles [4]. Especially, the influence of particle shapes and size as well as the size distribution requires a more physical approach, i.e., beds of particles as discrete phases, to give a better understanding of the transport phenomenon within packed beds.

Among numerical methods to computationally generate randomly packed beds of particles, the discrete element method (DEM), based on the original work of Cundall and Strack is the most popular one [2]. DEM is a numerical modeling approach to simulate the behavior of discrete systems where the interactions between individual particles are considered for the dynamic behavior of the system [5]. It makes it possible to model the

flow of fluids through packed beds and predict the performance of the porous media. The CFD-DEM coupling has also been recently used for evaluating the flow characteristics through the packed beds [6]. For instance, the CFD-DEM coupling has been widely used for studying and optimizing flow characteristics in fixed bed gasifiers [7], heat transfer in characteristics within packed beds of particles [8] and even including chemical reactions in particulate bed reactors [9].

For considering the particle-particle and particle-wall contact, Hertz-Mindlin contact model has been used [10] which is the most widely used approach for modeling the discrete phases. It takes into account both normal and tangential forces between particles, allowing for sliding or even rolling surfaces. On this regard, Lin et al. [11] discussed that the Hertz–Mindlin model could best describe a bed of under-compacted spherical rock particles.

There are several parameters that can influence the generation of a packed bed using DEM and a few studies in the literature have focused on parameters of bed generation with DEM in commercial software. Tangri et al. [5] has focused on the effect of particles drop height and intensity as well as friction and restitution coefficients on generation of packed beds of particles. They have also realized that the Hertz-Mindlin contact model could over-predict the packing density unless a proper rolling friction coefficient could be included. Moreover, Bester et al. [12] has studied the effect of Young's modulus of particles on bed generation.

One of the parameters that affects the accuracy of the DEM simulations for generating packed beds of particles is rolling friction between the particles and the particles and the wall. Rolling friction is a moment-induced to the contact point between two discrete elements that could mimic the rolling behavior of two complex shaped particles [13–15]. Goniva et al. [16] showed that the CFD-DEM simulations results would be improved to a great extent only by applying a proper rolling friction model. Wang et al. [17] found that by selecting proper rolling friction the DEM simulation with sphere particles can realistically represent the beds of non-spherical ones albeit finding the optimized value for this key parameter could be challenging. They have focused on the effect of rolling friction of particles on packing the beds and analyzed the effects of the rolling friction on the morphology and porosity using lignin particles. Furthermore, Soltanbeigi et al. [18] compared shape representations by exploring multiple aspects, including diverse rolling friction models and particle shapes, and showed that proper rolling friction models could help realistic representation of densely packed beds of granular beds.

In DEM simulations in commercial software, it is also crucial to find the optimum number of sub-steps for generating the packed beds. A sub-step is a smaller increment in time within the larger solver time step and helps the rapid changes of particles interactions be captured more accurately. During each sub-step, the DEM solver calculates the forces and interactions between particles, which means in each DEM sub-step, the positions and velocities of all particles are updated, and larger number of sub-steps would mean a more accurate acquisition of particles positions and velocities though resulting in higher computational times [19]. Thereby, smaller number of DEM sub-steps while generating the packed bed could lead to an under-resolution of the interactions between the particles, which can result in an inaccurate simulation of the packed bed and, consequently, an overestimation of the bed void fraction. However, larger numbers of sub-steps could increase the computational time and cost of DEM simulations.

When considering the bulk density and void fraction of packed beds in practice, two distinct modes of packing, loosely and densely packed, exist. Loosely packed beds represent arbitrarily formed beds with the gravity working on particles, while densely packed beds are formed after loosely packed beds are compacted further with vibration or external forces. In the experiments explained in Ref. [20], the crucible was lightly tapped 10 times on the table, following European Biochar Certificate recommendation [21]. Meanwhile, ISO 17828:2015 (Solid biofuels—Determination of bulk density) [22] recommends three shock exposures by dropping the container from 150 mm of height. The difference between the densely packed beds and loosely packed ones is the extent to which the particles fill the

gaps and void fraction becomes smaller. In case of the densely packed beds, the contact surface area between the particles increases and the heat and mass transfer could be facilitated compared to loosely packed beds. In order for the DEM-generated packed beds to describe the reality of densely packed beds and to estimate packing properties (e.g., void fraction) accurately, a specific set of numerical parameters shall be utilized in DEM simulations, such as rolling friction, Young's modulus, and time steps. Moreover, it is important to elucidate if the shape of particles would affect the representation of packed beds. However, no study could be found in the literature that has studied the role of these parameters to accurately describe densely packed beds consisting of densified, irregular-shaped biochar particles.

The objective of this study is to investigate the influence of independent parameters used in DEM on realistically generating the densely packed beds. The sensitivity of the generated packed beds void fraction has been analyzed based on different rolling friction values and the numbers of sub-step for conditions similar to experiments mentioned in [20]. STAR CCM+ (developed by Siemens Digital Industries Software, version 2310 (18.06.006-R8)) was used for the implementation of DEM simulations and the workflow has been validated for a wide range of particle sizes. Moreover, the influence of particles shapes has been investigated on the void fraction of packed beds in the form of particles with different aspect ratios. In this regard, the other novelty of this study would revolve around application of proper rolling friction coefficients on spherical particles that could avoid large computational costs and times for simulating industrial packed beds of complex-shaped particles.

## 2. Materials and Methods

### 2.1. Discrete Element Method (DEM)

In this study, STAR-CCM+ is employed for generation of packed beds of char particles using Discrete Element Method (DEM). The DEM solver in STAR-CCM+ employs a time-stepping algorithm to solve the equations of motion for each particle in the domain [23].

This contact model is based on the Hertz-Mindlin contact theory [24] where the theories of Hertz and Mindlin are used to model the normal and the tangential force-displacement relationships [5].

According to this model, the contact forces between elastic particles *A* and *B* could be described by [23]:

$$F_{contact} = F_n \boldsymbol{n} + F_t \boldsymbol{t}, \tag{1}$$

where $F_n$ and $F_t$ are the normal and tangential components of the contact force, and $\boldsymbol{n}$ and $\boldsymbol{t}$ are the vectors normal and tangential to the contact surface. The normal component could be defined as:

$$F_n = -K_n d_n - N_n v_n, \tag{2}$$

where $d_n$ is the particle overlaps in the directions normal to the contact point and $v_n$ is slip velocity of the contact point. $K_n$ and $N_n$ are normal spring stiffness and normal damping, respectively and can be obtained by:

$$K_n = \frac{4}{3} E_{eq} \sqrt{d_n R_{eq}}, \tag{3}$$

and,

$$N_n = \sqrt{(5 K_n M_{eq})} N_{n,damping}, \tag{4}$$

$M_{eq}$, $R_{eq}$ and $E_{eq}$ are the equivalent particle mass, radius, and Young's modulus, respectively and can be obtained as:

$$M_{eq} = \frac{1}{\frac{1}{M_A} + \frac{1}{M_B}}, \tag{5}$$

$$R_{eq} = \frac{1}{\frac{1}{R_A} + \frac{1}{R_B}}, \tag{6}$$

and,

$$E_{eq} = \frac{1}{\frac{1-\nu_A^2}{E_A} + \frac{1-\nu_B^2}{E_B}}, \tag{7}$$

where $M_A$ and $M_B$ are masses of particles $A$ and $B$, $R_A$ and $R_B$ are the radii of the particles, $E_A$ and $E_B$ are the Young's modulii of the particles and $\nu_A$ and $\nu_B$ are the Poisson's ratios.

$N_{n,damping}$ is the normal damping coefficient and is calculated by:

$$N_{n,damping} = \frac{-\ln(C_{n,rest})}{\sqrt{\pi^2 + \ln(C_{n,rest})^2}} \tag{8}$$

Here, $C_{n,rest}$ is the normal coefficient of restitution, which refers to the characteristics of collision behavior of particles. When particles collide in DEM simulations, this parameter determines the amount of energy lost or gained during the collision. The values of normal and tangential restitution coefficients typically range between 0 (perfectly inelastic collision) and 1 (perfectly elastic collision) and the values are between 0 and 1 for most materials in practice, which indicates a partially elastic collision where some energy is dissipated.

Similar to the normal component of contact force, the tangential component related to the surface friction between the particles could be obtained as:

$$F_t = \min\left(-K_t d_t - N_t \nu_t, \frac{|K_n d_n| C_{fs} d_t}{|d_t|}\right), \tag{9}$$

where $C_{fs}$ is the static friction coefficient, $\nu_t$ is slip velocity of the contact point, and $d_t$ is the length of particle overlap in the directions tangential to the contact point.

Similar to normal force components, the tangential spring stiffness, $K_t$, and the tangential damping, $N_t$, could be obtained as:

$$K_t = 8G_{eq}\sqrt{d_t R_{eq}}, \tag{10}$$

and,

$$N_t = \sqrt{(5K_t M_{eq})} N_{t,damping}, \tag{11}$$

where $N_{t,damping}$ is the tangential damping coefficient and $G_{eq}$ is the equivalent shear modulus. They could be obtained as:

$$N_{t,damping} = \frac{-\ln(C_{t,rest})}{\sqrt{\pi^2 + \ln(C_{t,rest})^2}} \tag{12}$$

and,

$$G_{eq} = \frac{1}{\frac{2(2-\nu_A)(1+\nu_A)}{E_A} + \frac{2(2-\nu_B)(1+\nu_B)}{E_B}}, \tag{13}$$

where $C_{t,rest}$ is the tangential restitution coefficient. To consider the particle-wall interaction, the wall radius and mass should be assumed to be $R_{wall} = M_{wall} = \infty$; therefore, the equivalent radius and mass will be $R_{eq} = R_{particle}$ and $M_{eq} = M_{particle}$, respectively.

In the case $-K_t d_t - N_t \nu_t > \frac{|K_n d_n| c_{fs} d_t}{|d_t|}$, and the particles have already started sliding on each other, the tangential component of the contact force is then defined by $F_t = \frac{|K_n d_n| \mu_r d_t}{|d_t|}$, and the static friction coefficient is then replaced by the rolling friction coefficient, $\mu_r$ [25].

As apparent from the equations above, the normal component of the contact force, $F_n$, is dependent on particle properties ($M_{eq}$, $R_{eq}$, $E_{eq}$ and $C_{n,rest}$) and temporary relative velocity and position ($d_n$ and $\nu_n$). Among these parameters, $E_{eq}$ and $C_{n,rest}$ are the independent

parameters related to the material properties that are common for all the particles in the simulation domain. Similarly, the tangential component of the contact force, $F_t$, is dependent on the material properties, $G_{eq}$, $C_{t,rest}$ and $\mu_r$, among which $G_{eq}$ and $C_{t,rest}$ are dependent to material properties. Therefore, the results of the DEM simulation, apart from case specific parameters such as density, particle size distribution, should be highly dependent on these parameters, together with the number of sub-steps. Meanwhile, these parameters are difficult to estimate based on first principle approaches, which motivate the exploration of these parameters in this study.

### 2.2. Simulation Conditions

Three different packed beds have been analyzed in this study with particles sizes covering a wide range of diameters from 315 μm to 6 mm. Table 1 specifies the minimum and maximum particle size and the parameters for the log-normal distribution of particle size followed by the distribution graph in Figure 1.

**Table 1.** Particles size distribution of the samples used in this study (parameters are based on log-normal distribution based on particle mass).

| Sample | Dmin [μm] | Dmax [μm] | Dmean [μm] | Standard Deviation [-] |
|--------|-----------|-----------|------------|------------------------|
| Dp3 | 315 | 400 | 355.6 | 0.0217 |
| Dp6 | 2000 | 3150 | 2526 | 0.293 |
| Dp7 | 4000 | 6300 | 5053 | 0.587 |

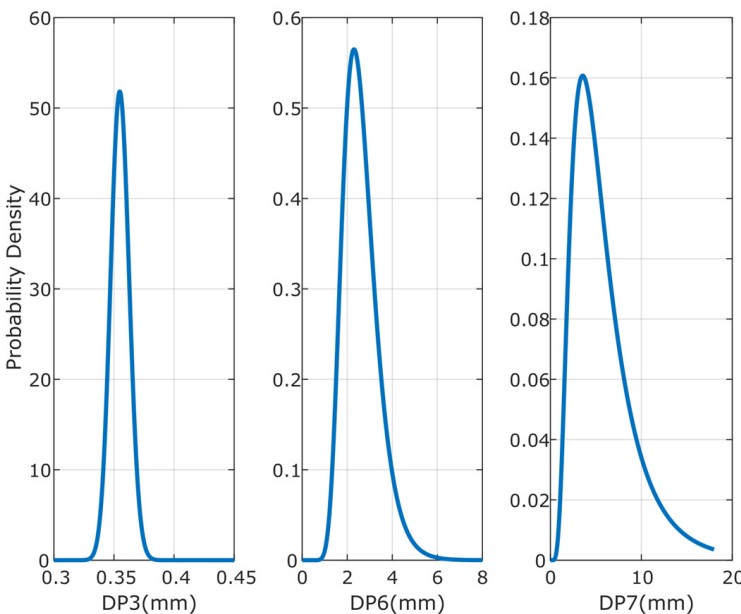

**Figure 1.** Log-normal distribution of the particle sizes in three studied samples: DP3 (**Left**), DP6 (**Middle**) and DP7 (**Right**).

The biomass char employed as the packed bed material is from dried spruce chips [20] and its intrinsic properties have been mentioned in Table 2. As mentioned earlier, Young's modulus and Poisson's ratio of biochar particles are considered as intrinsic properties of the material for which a range of experimental values exist in the literature and the values mentioned in Table 2 has been selected according to the properties of the respective biochar. However, it is also worth mentioning that the generated packed beds have not exhibited any significance variance in the void fraction or densification upon the sensitivity study applying different values of Young's modulus and Poisson's ratio within the range available in the literature (see supplementary Figures S1 and S2).

**Table 2.** Properties of the char particles.

| Property | Value Used in This Work |
|---|---|
| Particle Apparent Density | 783 kg m$^{-3}$ [20] |
| Young's Modulus | 6.8 GPa [26] |
| Poisson's Ratio | 0.33 [27] |

In addition, the normal and tangential restitution coefficients ($C_{n,rest}$, $C_{t,rest}$) have been 0.3, according to the shape of particles, the dropping angle of particles in the crucible and the material properties [28].

The geometry of the bed has been selected according to [13], which is a cylindrical crucible with diameter of 20 mm and height of 18 mm in order to validate the numerical data.

In order to capture the particle-particle interactions more accurately, the mesh size used in DEM should be as large as the largest particle, which will be coarser than typical CFD simulations. This is because the DEM solver models each particle as an individual entity and the performance of the bed is highly dependent on the size and shape of particles. If a large particle is assigned to a cell that is smaller than its diameter, it can lead to higher void fractions in the simulation than the reality. Large particles might not have enough space to properly interact with its neighboring particles, leading to a distorted representation of the packing arrangement [23].

For the samples modeled in this study, the mesh has been selected to be polyhedral and the mesh base size has been 2 mm for the samples Dp6 and Dp7 and 0.5 mm for Dp3. It should be noted that the DEM mesh base size is for the purpose of generating the packed bed of particles. For further simulation of fluid flow or heat and mass transfer, the mesh size shall be adjusted to match the requirements from each application.

The surface injector has been implemented at the top surface of the crucible to introduce particles into the simulation domain. Particle injection settings, i.e., injection rate and time, have been adjusted by observing the DEM simulation. Injection rate has been set to the maximum physical limit that avoids overlaps among the particles at the injection surface. Meanwhile, injection time has been adjusted to make the final volume of the packed beds be equal to the one in Ref. [20], i.e., 6500 mm$^3$.

### 2.3. Calculation of the Void Fraction

The DEM-generated packed bed has been analyzed in terms of the bed-scale volume-averaged void fraction together with the local area-based void fraction. The general definition of the volume-averaged void fraction ($\bar{\varepsilon}_{bed}$) is the ratio of the volume space between the particles ($V_{void}$) to the overall volume of the bed containing the particles ($V_{bed}$). This could be expressed as:

$$\bar{\varepsilon}_{bed} = \frac{V_{void}}{V_{bed}}, \tag{14}$$

In addition, radial and vertical distributions of area-based void fractions have been computed. For the radial profile, cylindrical sectional planes have been taken starting from the center of the cylinder and extending towards the wall. The area-based void fraction has then been calculated at regular intervals of 0.2 mm along these section planes. Similarly, for the vertical profiles, cross section planes parallel to the base have been taken along the height of the cylinder, starting from the bottom upwards. The area-based void fraction has been computed at increments of 0.2 mm. Figure 2 displays a schematic of the section planes over which the radial and vertical distributions of void fraction have been computed. The definition of area-based volume fraction can be expressed with the area of respective plane ($A_{plane}$) and the area of void in the plane ($A_{void}$) as:

$$\varepsilon_{bed,area} = \frac{A_{void}}{A_{plane}} \tag{15}$$

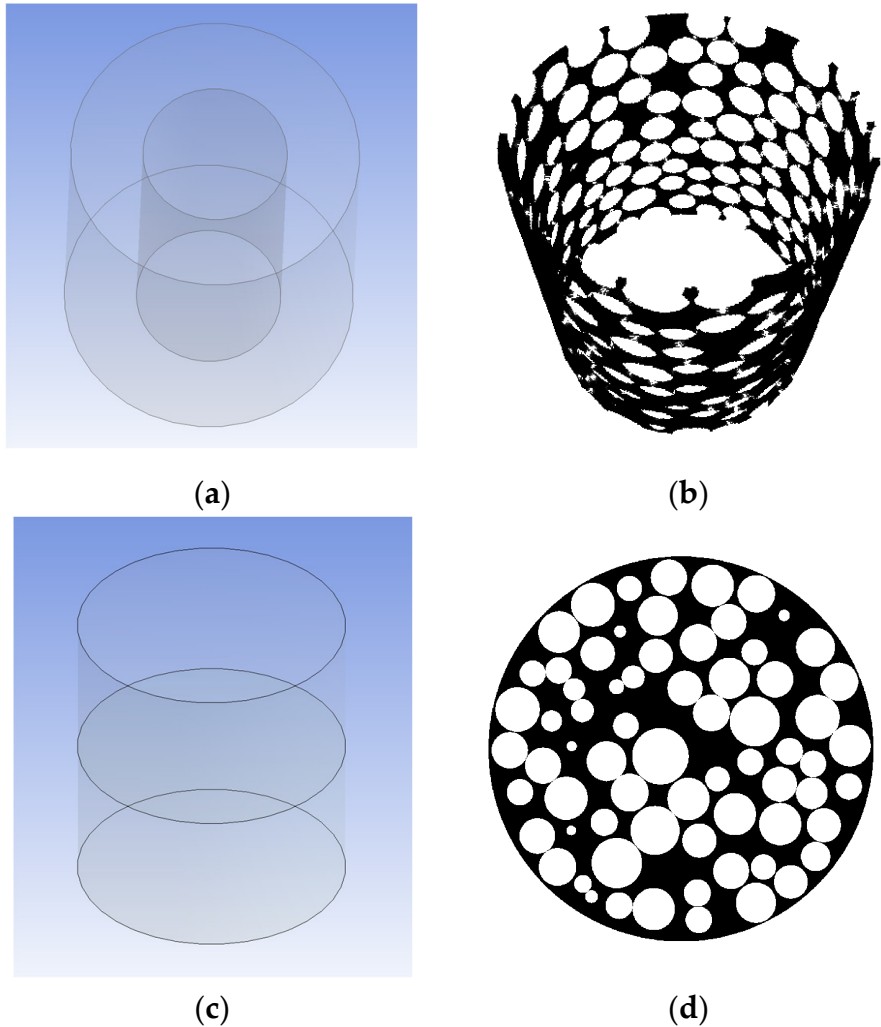

**Figure 2.** The schematic of section planes over which the area-based void fractions have been computed (**a**) along the radius and (**c**) along the height. The schematic of the pore space has been shown in the (**b**) radial section and (**d**) vertical one. White spherical areas depict particles and black areas represent the void between particles.

In STAR-CCM+, the "Packed Bed Void Fraction" is a field function used to calculate the void fraction in a packed bed of particles, defined as the ratio of the volume of the voids to the total volume of the packed bed [23]. This field function shall be accurate since it can be applied to further simulations of the packed bed and its correctness affects its overall performance. To get this built-in function in STAR-CCM+ as close to the real values as possible, the number of DEM sub-steps has significant influence. Using coarse sub-steps cannot capture the variation of void faction close to center of the cylinder and predicts the quite constant void fraction. This is due to the fact that coarser sub-steps in STAR-CCM+ indicates lower resolution of the DEM particles and consequently less capability of capturing the space between particles. It would be preferred to use smaller sub-steps but due to computational cost, especially for industrial applications where the bed size is larger, one shall find a trade-off between number of sub-steps and accuracy.

## 3. Results and Discussions

### 3.1. Validation of the Numerical Model for Different Particle Sizes

As mentioned earlier, the number of sub-steps is an important parameter in generation of realistic packed beds of particles in DEM method. To find the optimum value, different number of sub-steps have been used for generating packed beds and the void fraction

of the generated beds have been compared with the experimental values. These values have been determined as $1 - \frac{Bulk\ Density}{Envelope\ Density}$, where the bulk density is measured using the VDLUFA-Method A 13.2.1 as the ratio of mass to volume of the packed bed of particles, and the envelope density by immersing the samples in $Al_2O_3$ in a graduated cylinder (For more details, refer to [20]). For this step, sample DP6 (characteristics mentioned in Table 2) has been employed. The analysis has been implemented for the surface-averaged void fraction on the mid- plane for the sample mentioned in the materials and method section. Figure 3 presents the bed void fraction for different number of sub-steps.

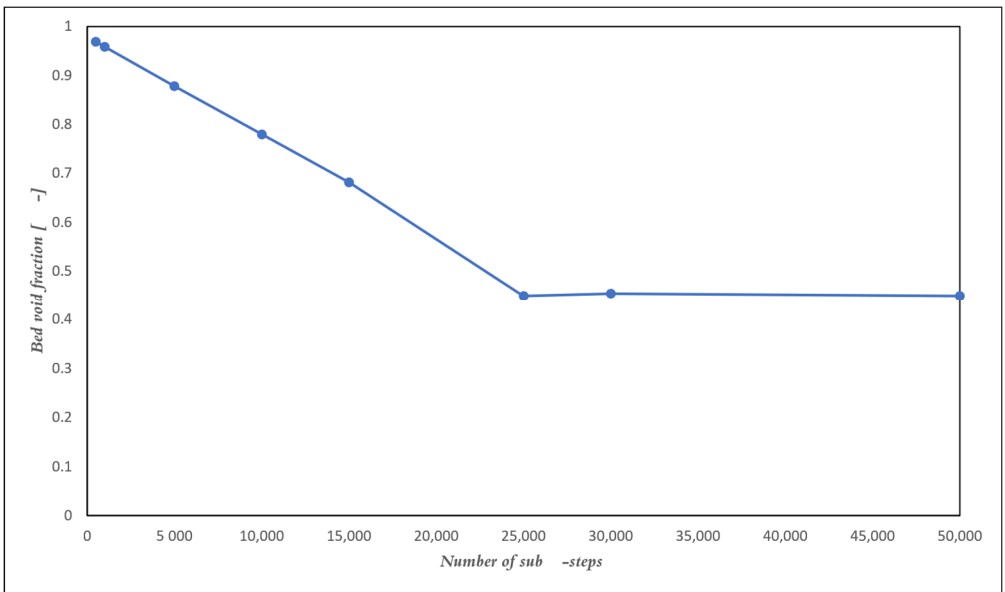

**Figure 3.** Bed void fraction versus the number of DEM sub-steps.

According to Figure 3, the minimum number of sub-steps is 25,000 since the bed void fraction is constant for sub-steps larger than 25,000. The larger the number of sub-steps is, the higher the simulation time and cost are. For the small-scale application of this study, it might not be significant to lower the time and cost; however, it is crucial for industrial and large-scale applications to make a trade-off between the accuracy of the results and the computational time and cost. It should be noted that similar DEM sub-step analysis shall be conducted for beds with much larger or much smaller particles; however, similar value for number of sub-steps has been obtained for all the other samples in this study. Table 3 presents the computational time and cost for generating packed beds with different number of sub-steps. It shall be noted that all simulations have been running with 16 CPU of Intel(R) Xeon(R) Silver (2.10 GHz) and 144 GB of DDR4 RAM memory. For large scale problems, it is recommended that the injection and packing of the bed be done in smallest number of sub-steps possible (e.g., 500 or 1000) and after the bed is completely packed, simulation runs for some time with the optimum number of sub-steps to obtain the realistic value of the void fraction and the built-in function of the void fraction in STAR-CCM+. As of this combination, the computational time decreased to 350 min, which is almost 10 times smaller than running the whole simulation with 25,000 sub-steps from the beginning, while the bed void fraction ends up the same value.

**Table 3.** Computational time for packed bed generation with respect to number of sub-steps.

| Number of Sub-Steps | Computational Time (min) | Bed Void Fraction |
| --- | --- | --- |
| 5000 | 90 | 0.88 |
| 10,000 | 200 | 0.78 |
| 15,000 | 320 | 0.62 |
| 25,000 | 3500 | 0.38 |

Using the abovementioned approach for the other two samples, DP3 and DP7, Figure 4 displays the comparison between the bed void obtained from experimental data and DEM-generated beds in STAR-CCM+. The samples have been selected the way that they could cover a wide range of particle sizes, which in turn represent a wide range of $R/d_p$ (3.9, 7.8, and 55.9).

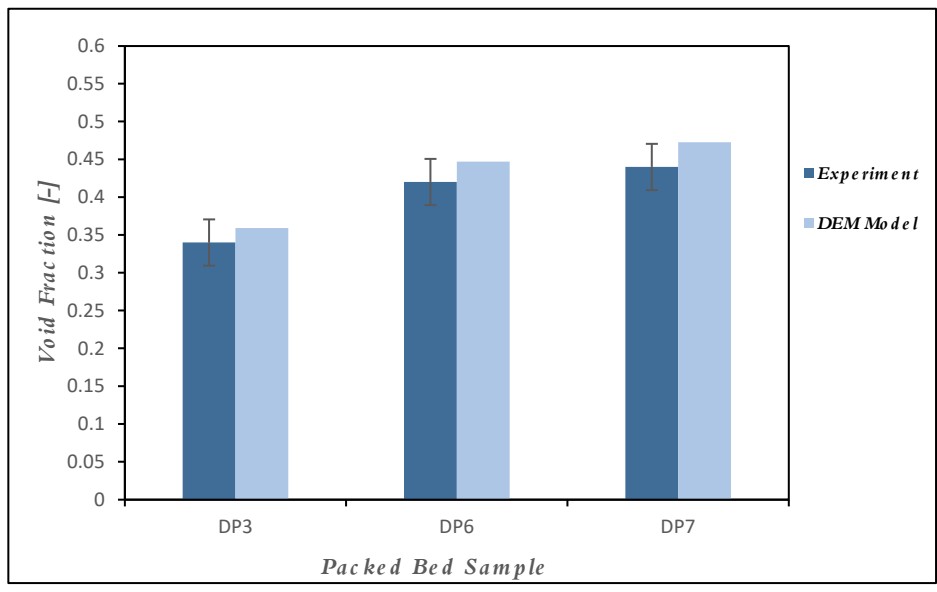

**Figure 4.** Comparison between the bed void fraction from the experiment and the numerical model.

According to Figure 4, the prediction of the volume-averaged bed void fraction by the DEM model has been in good agreement with the experimental data. The error range between 5% and 7% could be due to the fact that the real char particles are not perfectly sphere unlike in the DEM model, and this shape difference could affect the packing and the bed void fraction accordingly. Nevertheless, the DEM model correctly captured the effect of small vessel diameter (low $R/d_p$ ratio) as evident from the predicted trend (increase in volume fraction).

### 3.2. The Detailed Particle Distribution Inside the Packed Bed

Figure 5 shows a visualization of particles in the DEM-generated packed bed (a) and the void fraction distribution throughout the vertical middle plane of the domain (b). As Figure 5 presents, larger values of void fraction could be seen near the walls and at the top of the bed. At the top of the bed, the random injection of particles leads to some empty space above the last row of particles (as can be seen in Figure 5a) which leads to larger void fractions at the top of Figure 5b. Furthermore, close to the walls, the particles are aligned to the wall surface and inevitable gaps are then formed. This wall effect can be the cause of slightly higher void fractions near the walls and at the bottom of the crucible.

Figure 6 shows the radial distribution of void fraction along the cylinder. The x axis is the distance from the wall normalized by mean particle diameter, $(R - r)/d_p$, where $R$ is the radius of cylinder, $r$ is the radius of each cylindrical section and $d_p$ is the average particle diameter (2.5 mm in case of DP6). The void fraction values far from the wall (larger values of $(R - r)/d_p$) fluctuate slightly around 0.4 (close to the experimental value of the volume based void fraction) and the fluctuation increases as approaching the wall (Ratios closer to zero). Since the particles are aligned adjacent to the walls, the first large porosity drop near the wall could be observed at $(R - r)/d_p = 0.5$, which is equal to half of the mean particle diameter.

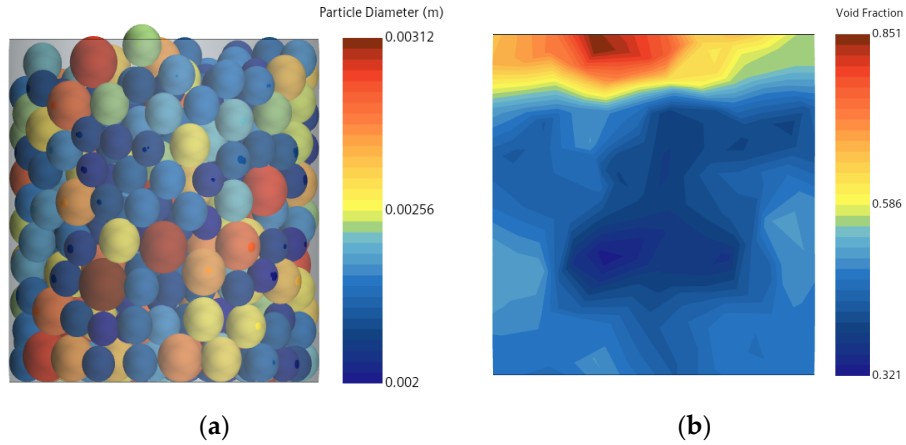

**(a)**          **(b)**

**Figure 5.** The DEM-generated packed bed (**a**) and the void fraction distribution throughout the middle plane of the packed bed (**b**).

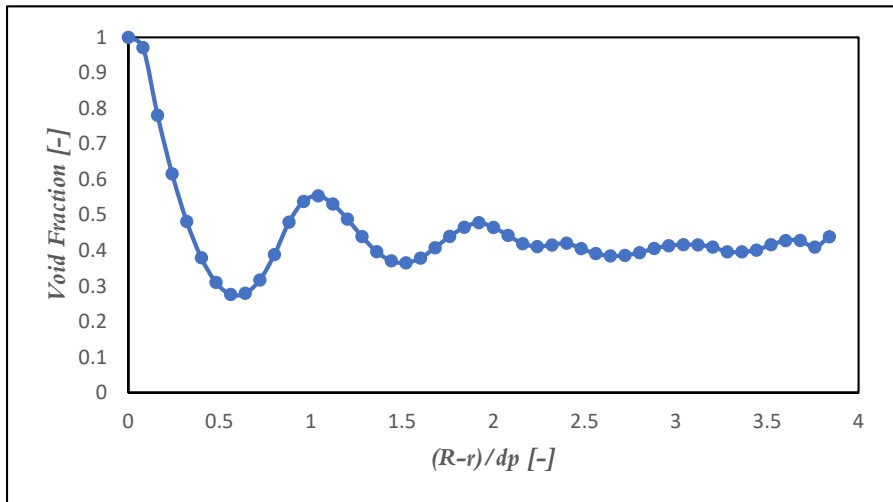

**Figure 6.** Radial distribution of the void fraction along radius of the packed bed.

Similarly, Figure 7 depicts the void fraction distribution along the height of the packed bed. To do so, the area-based void fraction has been calculated on the cross sections at every 0.2 mm distance from the bottom plane.

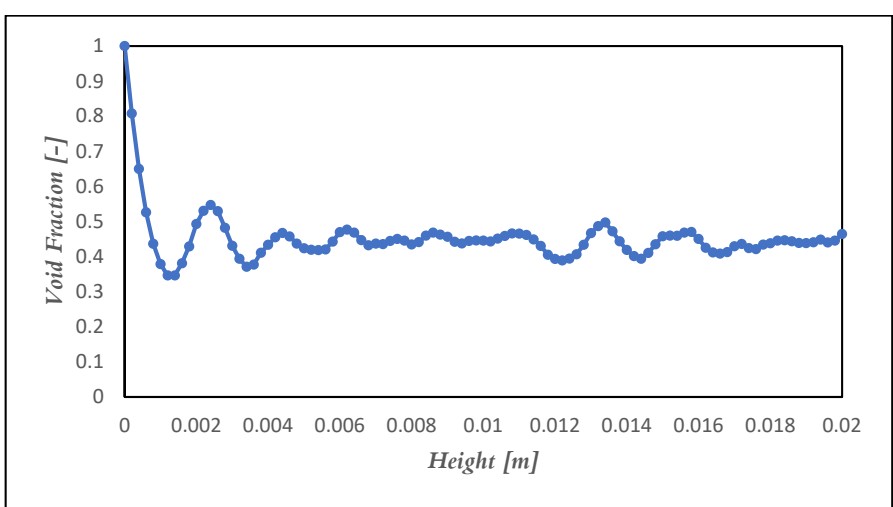

**Figure 7.** Vertical distribution of void fraction along the height of the packed bed.

At the bottom of the bed, void fraction was one due to the wall effect and the spherical shape of the particle. It is noticeable that the change of bed void fraction at the bottom of the bed is sharper, and this could be because the particles are aligned at the bottom. Up until 1.2–1.4 mm (half of the smallest particle diameter) the value of the void fraction goes down from unity to a minimum and then oscillates for a length of two particle diameters until it approaches towards the mean value. The sinusoidal behavior of the void fraction within heights 12 mm to 15 mm is possibly because the particles could have been structurally aligned in this region, albeit by coincidence.

As per Figures 6 and 7, it is apparent that the fluctuations of bed void fraction are considerable adjacent to the walls of the domain. For better understanding of the wall effects, the simulations have been carried out for a series of domains with $R/d_p$ ratios of 6, 8, 10, 12 and 16, among which the ratio equal to 8 is the original crucible size. Figure 8 demonstrates the void fraction distribution with respect to the normalized distance from crucible surface. It is observed that for the first half diameter of particle $(R - r)/d_p \approx 0.5$, the wall effect is similar except small difference which could be result of random filling of the pack bed. Regardless of the magnitudes of the $R/d_p$ ratios, the oscillation with the frequency of the mean particle diameter became insignificant after two particle diameters and was replaced by random fluctuation around mean void fraction. In other words, the inner domains would simulate the conditions of a densely packed bed.

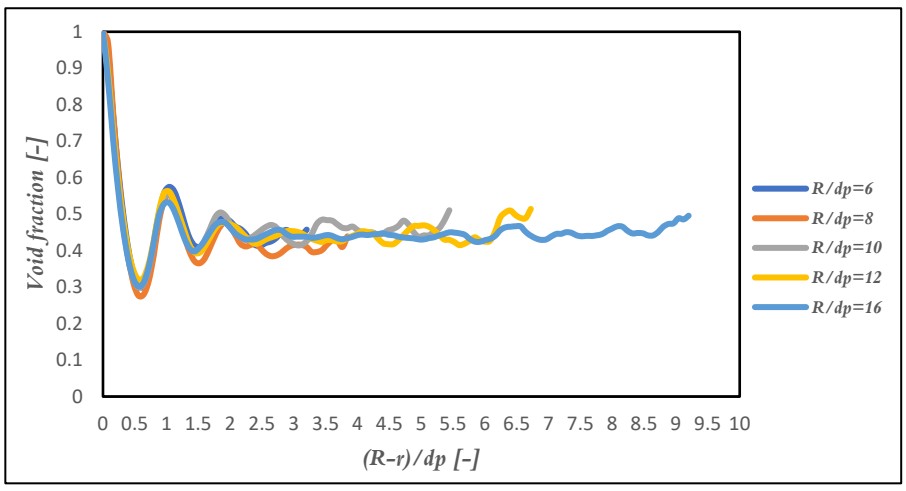

**Figure 8.** Radial distribution of the void fraction along radius of the packed beds with $R/d_p$ = 6, 8, 10, 12, 16, and 20.

### 3.3. Influence of Rolling Friction Coefficient on the Bed Void Fraction

As mentioned before, one of the crucial parameters in packing the particles is rolling friction between particles as well as the particles with the wall. For better understanding the role of rolling friction of particles, it is helpful to analogize it to the tapping of the container in order to redistribute the particles and decrease the void space between them. As mentioned before, the loosely and densely packed beds are different in sense of void fraction as well as from one material to another. Here, rolling friction coefficient shall be adjusted accordingly. Figure 9 demonstrates the volume-averaged void fraction of the packed beds in different values of rolling friction coefficient. Rolling friction coefficients smaller than $10^{-5}$ seem to represent the void fraction values close to the expected one for densely packed bed. At higher rolling friction coefficients, especially above 0.1, the void fraction increases along with rolling friction coefficients. The values become much larger than those for densely packed bed, and approach toward those for loosely packed bed. It is worth mentioning that, as generation of DEM-based packed beds is a random phenomenon, the sensitivity of the simulations to randomness has been checked by repeating the same simulation for three times and similar value of void fraction has been obtained.

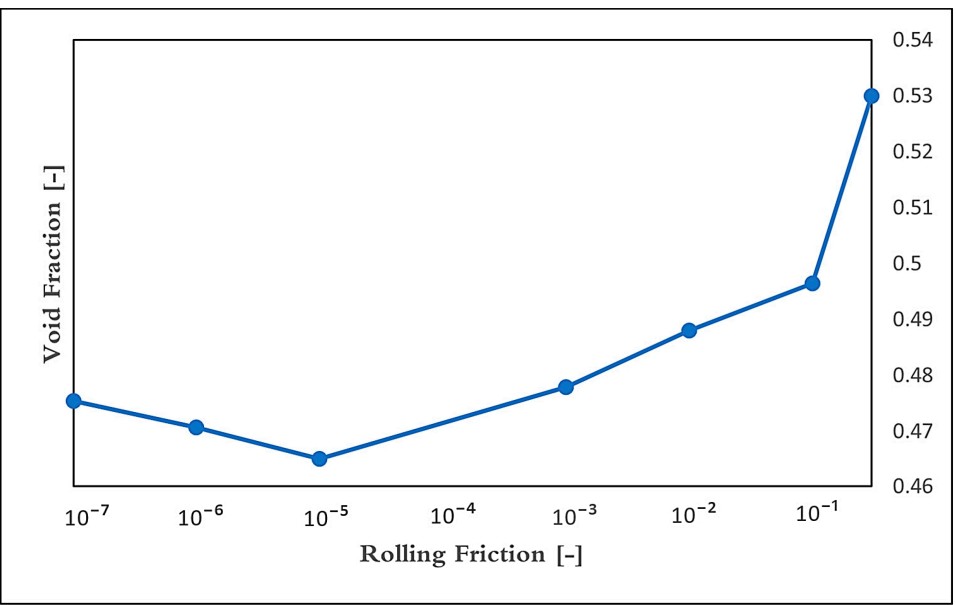

**Figure 9.** The volume-averaged void fraction of the beds generated with respect to different rolling friction coefficients.

In order to investigate the differences of packing in detail, Figures 10 and 11 show the influence of rolling friction on the void fraction distribution along the height of the packed bed and along the normalized distance from the wall, $(R - r)/d_p$, respectively. The void fraction has a variation and oscillation especially near the wall as discussed earlier. The void fraction has a variation and oscillation especially near the wall as discussed earlier. When comparing different rolling resistance coefficients, it is noticed that void fractions remain fairly consistent when the rolling resistance coefficient is smaller than 0.01. Moreover, the difference is more noticeable at the bottom of the packed bed than on the side walls. However, when we use larger rolling resistance coefficients, the simulations overestimate the values of void fractions compared to the experimental data. This overestimation gets more significant in vicinity of the walls.

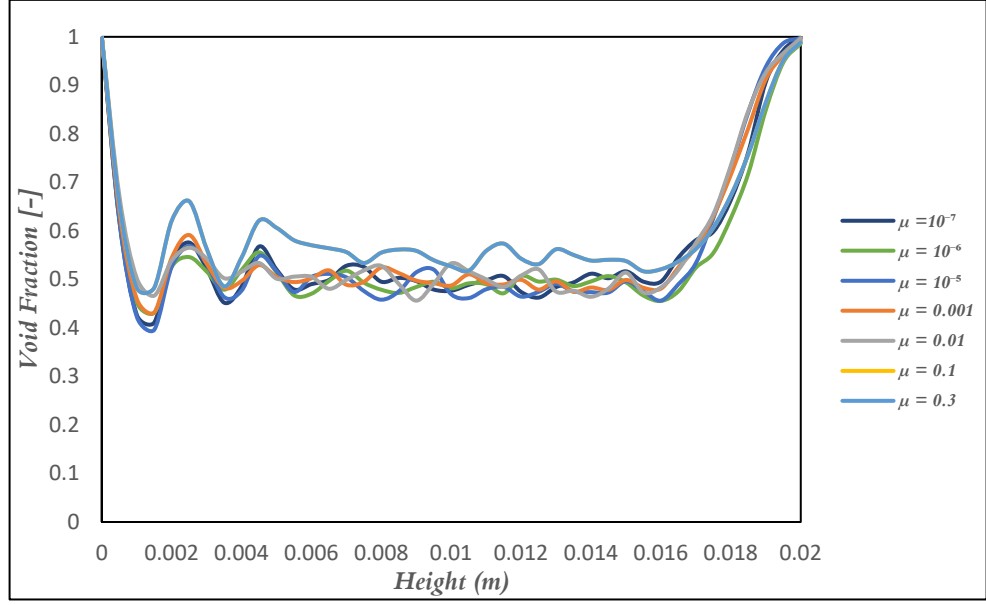

**Figure 10.** The influence of rolling friction coefficient of particles on the vertical distribution of void fraction.

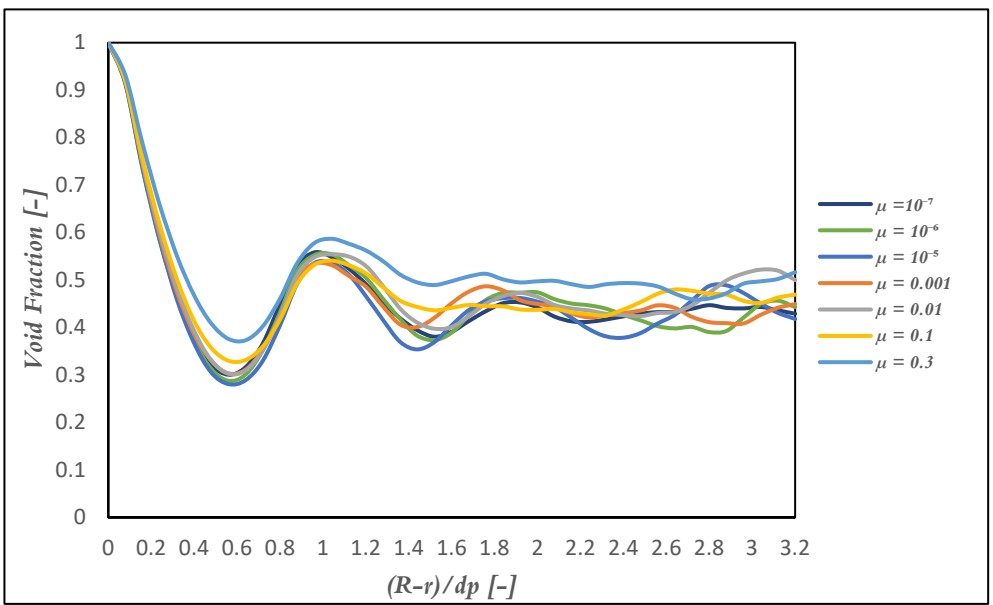

**Figure 11.** The influence of rolling friction coefficient of particles on the radial distribution of void fraction.

The results from the volume-averaged void fraction as well as the respective radial and vertical distributions (Figures 9–11) indicate that rolling friction coefficients below $10^{-3}$ are generally recommended for the DEM simulation of densely packed bed. Meanwhile, the void fraction of loosely packed bed is sensitive to the surface properties of materials and the selection of the rolling friction coefficients becomes important.to reproduce loosely packed bed accurately.

### 3.4. Influence of Particles Shapes on the Bed Void Fraction

Since the particles in practice deviate from being fully spherical, it is crucial to investigate the effect of shape factor on the void fraction. Hereby, the void fractions of the beds with the particle aspect ratios (length to diameter ratio) of 1.2, 1.5, 2, 2.2 and 3.5 have been assessed in this section for the sample DP6. The reason behind selecting the mentioned aspect ratios is the fact that the char particles, in reality, do not deviate much from spheres and ratios beyond these would not reflect the realistic char particles. Figure 12 shows the external views of the packed beds containing non-spherical particles. The particles with the aspect ratios up to 1.2 resemble sphere particles and it is hard to distinguish the particle orientation visually. The particles with the aspect ratios above 2 resemble cylindrical particles and clearly exhibit the alignment of particles. Visually, this particle elongation seems to result in the larger gaps between particles as expected. Particles with the aspect ratio of 1.5 show the transition behavior, that is, the particle orientation is clearly visible but the gaps between particles do not seem to be widened.

To quantitatively discuss the previous observation for the beds presented in Figure 12, the volume-averaged bed void fractions have been calculated according to equation (14). Figure 13 shows the changes in the void fraction for different beds as well as the value mentioned in the experimental results of [20] and the one obtained from the spherical particles (aspect ratio of 1). The yellow-shaded region in the graph represents the standard deviations of void fractions estimated based on the error propagation from the standard deviations of bulk and envelope density [20].

The void fractions of beds with non-spherical particles deviate from the experimental value of 0.42 and the one from spherical particles, i.e., 0.447 for DP6. It could be observed that aspect ratios of 1.1 up until 1.5 show deviation range of 0.03 to 0.06, yet still in the acceptable margin; however, for aspect ratios larger than 1.5, meaning particles shifting shape from spheres to cylinders, the bed void fractions are much larger than both the

experimental values and the values from beds with spherical particles. This could be due to the fact that the cylinder-shaped particles attain different orientations when being packed and the gap between them becomes more than densely packed spherical particles. It should be noted that this result could depend on type of particles, and, in this study, one could see that packed beds of spherical particles could exhibit the same range of void fractions as the experimental data of char particles.

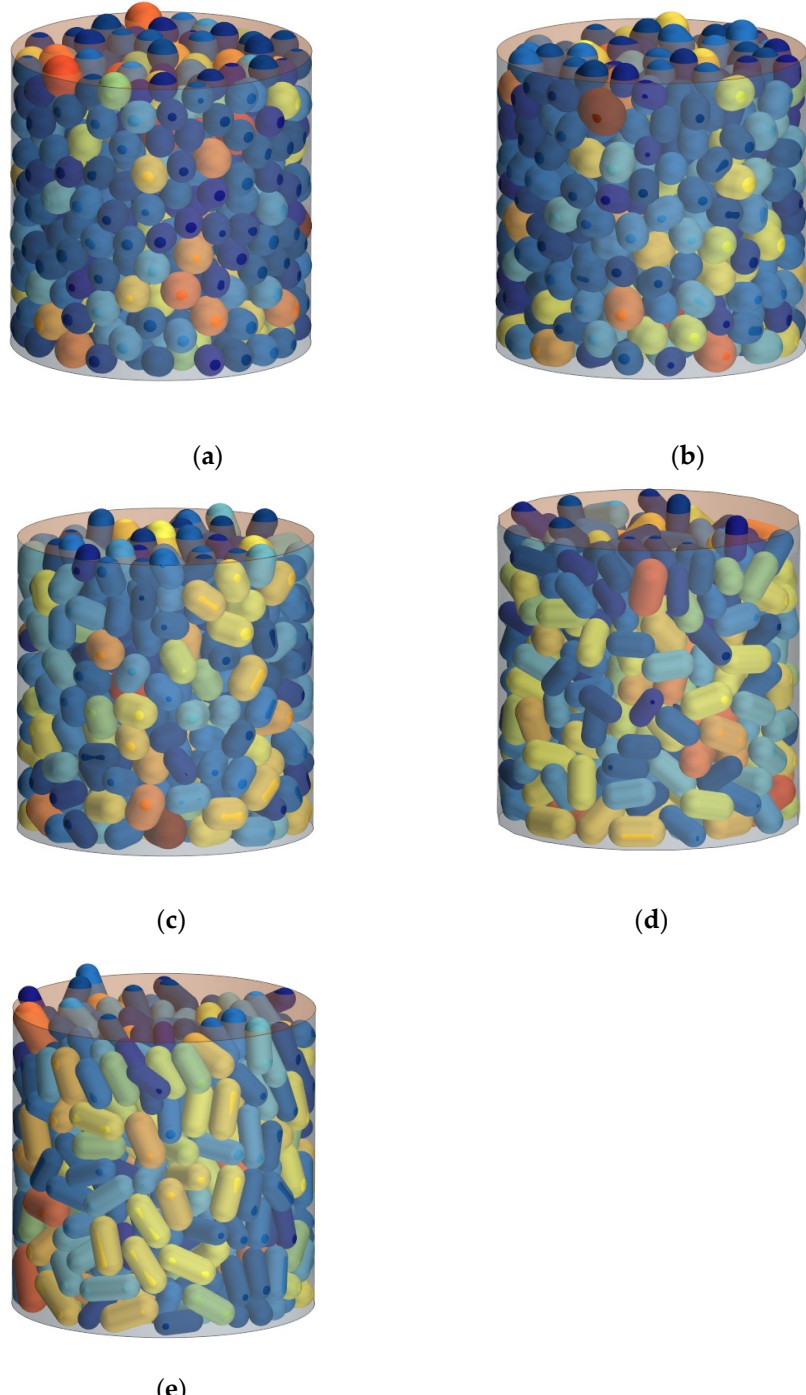

**Figure 12.** External views of packed beds of non-spherical particles with aspect ratios of (**a**) 1.2, (**b**) 1.5, (**c**) 2, (**d**) 2.2, and (**e**) 3.5. Different colors are for the purpose of better visualization and carry no value significance.

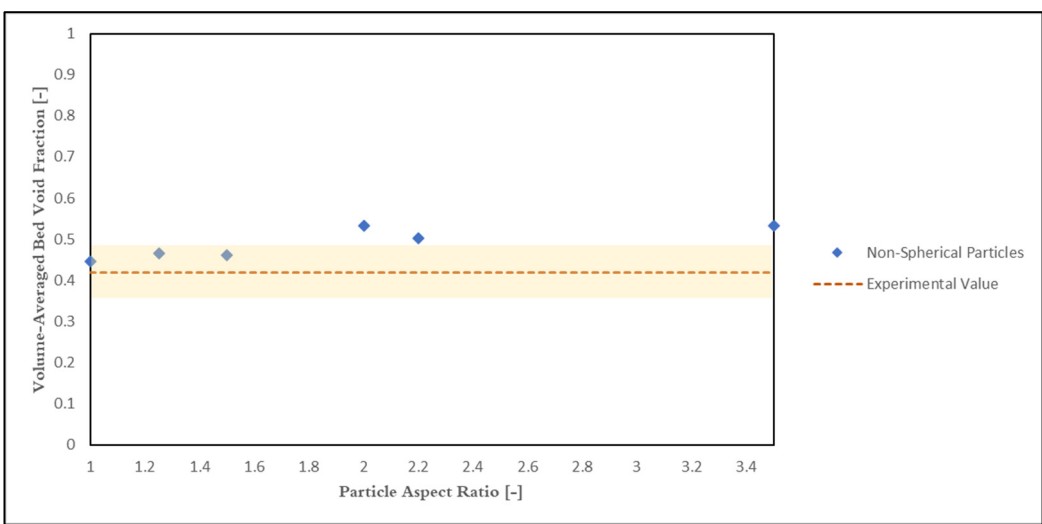

**Figure 13.** Volume-averaged bed void fraction for different particle aspect ratios for DP6 sample.

## 4. Conclusions

In this investigation, the DEM (Discrete Element Method) approach, implemented within STAR-CCM+, was employed to generate numerical models of packed beds with spherical particles.

The initial section of this study evaluated how the chosen DEM method for generating packed beds influences porosity. This assessment concludes that the number of sub-steps used should be adequately large to capture particle resolution yet not excessive to avoid excessive computational time and cost. For a sample size range of 2 mm to 3 mm, an optimal number was identified through a trade-off approach. A suggestion was made to employ a smaller number of sub-steps during particle injection, gradually increasing to a higher value for settling. The generated bed has exhibited good agreement with the experimental results with the same geometry and particle size range. Moreover, a detailed analysis of the void fraction distribution in radial and vertical directions revealed that inner cylinder regions closely aligned with experimental values, while nearer to the wall, fluctuations led to greater void fraction variations.

The second part of the study investigated the impact of rolling friction coefficient, both among particles and with the wall, on the precision of packed bed generation. It was determined that rolling friction coefficients exceeding $10^{-5}$ resulted in significant deviations from the void fraction of densely packed bed, while coefficients below this threshold led to minor deviations from experimental values. Moreover, it is concluded that choosing a proper rolling friction at higher values could create similar conditions as of a loosely packed bed albeit the outcome would be affected by material properties and packing conditions significantly. Consequently, selecting an appropriate rolling friction value becomes pivotal for achieving accurate simulations of particle packed beds using the DEM method.

In the last section of this study, the influence of particles morphology has been investigated. Six different ellipsoidal shapes with aspects ratios of 1.2, 1.5, 2, 2.2 and 3.5 have been studied and the volume-averaged bed void fraction for one of the samples have been compared with spherical and experimental values. The results showed that deviations from spherical towards cylindrical-shaped particles increase the void fraction by more than 0.1. However, this deviation was not significant up to the aspect ratios of 1.5 and applying suitable rolling friction coefficients for beds of spherical particles could well mimic the behavior of non-spherical particle.

**Supplementary Materials:** The following supporting information can be downloaded at: https://www.mdpi.com/article/10.3390/pr12010183/s1, Figure S1: Vertical distribution of void fraction for different Young's modulus numbers, Figure S2: Vertical distribution of void fraction for different Poisson ratios.

**Author Contributions:** Conceptualization, Z.G.M.; methodology, Z.G.M.; software, Z.G.M.; validation, Z.G.M. and K.U.; formal analysis, Z.G.M.; investigation, Z.G.M.; resources, K.U.; writing—original draft preparation, Z.G.M.; writing—review and editing, K.U. and J.G.I.H.; visualization, Z.G.M.; supervision, K.U. and J.G.I.H.; project administration, K.U.; funding acquisition, K.U. All authors have read and agreed to the published version of the manuscript.

**Funding:** This study was funded by Swedish Energy Agency (Project number: P46974-1), part of which is funded by The Recovery and Resilience Facility, RRF of European Commission (EU).

**Data Availability Statement:** Data are contained within the article.

**Acknowledgments:** We express our gratitude to Muhammad Aqib Chishty for his contribution that helped the quality of this paper.

**Conflicts of Interest:** The authors declare no conflict of interest.

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
