# Peer review of "The Impact of Discrete Element Method Parameters on Realistic Representation of Spherical Particles in a Packed Bed"

_processes, doi:10.3390/pr12010183_

Round 1
Reviewer 1 Report (New Reviewer)
Comments and Suggestions for Authors
Dear Editor,
I have read the paper “The impact of DEM (Discrete Element Method) Parameters on Realistic Representation of Spherical Particles in a Packed Bed”. I suggest that the paper should be revised accordingly. The detailed comments are listed below.
1. Title of the paper: It is unconventional to have “DEM (Discrete Element Method)” in the title. I will suggest the authors to use “Discrete Element Method” directly. In the title, the authors stated that this paper is for a realistic representation of spherical particles, but there are investigations on non-spherical particles. Thus, I think the title can be more general.
2. Abstract, Line 12: change “axial” to “radial”
3. Abstract, Line 18-19: The last sentence shows a very specific result, which is not necessary for an abstract.
4. Page 2, Line 64-65: The authors stated that, with citation of Ref. 10, rolling friction is a momentum …. However, Ref. 10 stated that rolling friction is a moment. There are huge differences between a momentum and a moment. Their units are totally different. The authors should clarify this and make sure that they specified correct parameters.
5. Page 3, Line 111: delete “Packed Bed Generation by Using”
6. Page 3, Line 128: Change “Req” to “Req”.
7. I did not see the authors provide specific values for the static frictional coefficient. The value of rolling resistance should be mentioned in the methodology part.
8. Page 4, Line 133: change “modulus” to “moduli”, since moduli is the plural form of modulus.
9. The authors did not describe how the sub-step functions in the numerical procedure.
10. Page 8, Figure 3: The curve changes from decreasing to being flat as we increase the x axis. Is this sudden change in slope universal? In other words, will the transition point change if we change either the particle sizes, shapes, stiffness, or frictions?
11. Page 10, Figure 7: in this figure, within Height = 0.011~0.015, there is a sine function-like curve. Why is it showing such a behavior? Is it because that particles in this range are also aligned structurally?
12. Page 12, Figure 9: In this figure, for each data point, how many DEM simulations have been performed? Is only one simulation per data point enough to rule out the randomness of DEM simulations?
13. Page 12, Figure 10: The legend of this figure shows values of R, but R is not the rolling resistance.
14. Page 13, Figure 11: Same as the last comment. The legend is not consistent with the figure caption.
Author Response
Please see the attachment.

Reviewer 2 Report (New Reviewer)
Comments and Suggestions for Authors
Paper is in good shape and to investigate the effects of DEM parameters Realistic Representation of Spherical Particles in a Packed Bed. I am going to accept the paper after considering the following comments.
1) The number of discussed papers in the introduction is not sufficient. Please discuss more recent papers.
2) How experimental data in Figure 4 are obtained? Please explain how experiments have been conducted? Is it the one that has been explained in Section 2.3? this matter is not clear.
3) Please emphasize the novelty of the work in your paper. What is the most important significant contribution?
Round 2
Reviewer 2 Report (New Reviewer)
Comments and Suggestions for Authors
I am going to accept the paper as the comments were answered properly.
This manuscript is a resubmission of an earlier submission. The following is a list of the peer review reports and author responses from that submission.
Round 1
Reviewer 1 Report
Comments and Suggestions for Authors
1. Equations (5) and (6) are the same. Please correct.
2. Is there any reason that apart of the 0.35 mm packed bed, only 6- and 7-mm particles has been chosen? Please justify.
3. No information regarding the type of material used as particles in this study is provided.
4. Given the computation complexity of this study, the RAM memory along with the CPU characteristics (type, frequency) should be given.
5. No significant differences in terms of rolling friction coefficient is shown in Figure 10. Please explain.
Reviewer 2 Report
Comments and Suggestions for Authors
Thank you for your paper!
I found it clear and easy to understand throughout.
The Introduction gives a good description of the background.
The method is described well.
The results show the sensitivity of void fraction to rolling friction coefficient. This will be of great use to those setting up packed bed simulations with DEM.
I suggest only some minor edits:
Line 73 "softwares" should be "software"
Line 90 "exposures"
Line 102 "rolling frictions" should be "rolling friction values"
Eq 6 appears to be a duplicate of Eq 5
Line 164 "principal" should be "principle"
Comments on the Quality of English Language
Please see my "Comments and suggestions for Authors"
Reviewer 3 Report
Comments and Suggestions for Authors
Overall, it is a poor contribution on the field of simulation of packed bed reactors using the discrete element method. Obviously, the study lacks novelty since the study of packed bed reactors with spherical particles is not something new. I suggest the following improvements
[1] the study should account for the effect of the particle shape (e.g., ellipsoidal particles)
[2] the study should account for the effect of the flow of the surrounding fluid
